# NDAD: Negative-Direction Aware Decoding for Large Language Models via Controllable Hallucination Signal Injection

**Panjia Qiu**[1] **Mingyuan Fan**[1] **Cen Chen**[1*] **Daixin Wang**[2]

School of Data Science and Engineering, East China Normal University    [2]AntGroup

panjiaqiu@stu.ecnu.edu.cn    fmy2660966@gmail.com

cenchen@dase.ecnu.edu.cn    daixin.wdx@antgroup.com

## ABSTRACT

Large language models (LLMs) have recently achieved impressive progress in knowledge-intensive and reasoning tasks. However, their tendency to produce fabricated or factually inconsistent content remains a fundamental challenge to their practical deployment. To address this issue, we propose Negative-Direction Aware Decoding (NDAD), a novel decoding method that identifies and exploits hallucination signals as repulsive directions in the model's representation space, thereby improving factual adherence without retraining. Specifically, NDAD elicits hallucination-leaning signals by selectively masking critical attention heads, which exposes unstable hypotheses that the model would otherwise amplify during generation. To regulate the influence of these signals, NDAD employs two complementary weights: a global alignment weight measuring how well the induced signal aligns with the layer's native activations (thus quantifying its referential utility) and a local weight estimating whether low-probability tokens in the masked distribution are likely to evolve toward the final output. Based on the weights, we derive a latent hallucination distribution that serves as the negative direction. A lightweight gradient-descent step then subtracts mass from hallucination-prone regions of the output distribution, adjusting the final logits while preserving the model's high-confidence predictions. Extensive experiments across multiple LLMs and diverse benchmark datasets demonstrate that NDAD consistently enhances factual reliability without requiring additional training or external knowledge.

## 1 INTRODUCTION

In recent years, large language models (LLMs) have achieved remarkable breakthroughs over various tasks (Achiam et al., 2023; Anil et al., 2023; Touvron et al., 2023b;a; Team et al., 2023). However, a pervasive challenge is the phenomenon of hallucination, wherein LLMs generate factually incorrect, fabricated, or nonsensical information with high confidence (Zhang et al., 2023b; Li et al., 2024; Fan et al., 2025).

Current approaches to mitigate hallucinations mainly fall into two categories: retrieval-augmented methods (Li et al., 2023b; Min et al., 2023) and training-based methods (Tian et al., 2023; Rafailov et al., 2023). Retrieval-augmented methods, while effective, often introduce architectural complexity, latency, and dependency on the availability and integrity of external large-scale databases. Training-based methods, on the other hand, can be computationally intensive and may struggle to generalize across diverse factual domains. A less explored, yet highly promising, avenue is the optimization of the decoding process itself (Welleck et al., 2024; Shi et al., 2024). Importantly, prior studies suggest that LLMs already encode factual signals within their internal representations as a byproduct of large-scale pretraining, though conventional decoding techniques often fail to surface this latent knowledge (Wang et al., 2020; Kadavath et al., 2022; Li et al., 2023a; Saunders et al., 2023). Motivated by this observation, intervention-based decoding methods (Chuang et al., 2023; Li et al., 2023a; 2022; Zhang et al., 2023a) have been developed to exploit these factual signals to alleviate hallucinations.

---

*Corresponding author.

In this study, we propose a novel yet quite effective intervention decoding method called Negative-Direction Aware Decoding (NDAD). Unlike prior approaches that focus on extracting latent factual cues from early layers, NDAD instead identifies hallucination signals and then leverages them to calibrate the decoding process. As shown in Figure 1, NDAD detects latent distributions that correlate with factually incorrect outputs by masking influential attention heads in the model. We then develop two complementary weights based on the detected distributions to suppress hallucination signals. Specifically, the global weighting component evaluates the alignment between hallucination-oriented logits and earlier-layer logits, estimating whether such trajectories reflect distributions the model is more likely to generate. In parallel, the local weighting component tracks high-risk tail tokens to assess their likelihood of advancing toward the final output. The final token selection is then guided by a single-step gradient-descent adjustment, which penalizes the generation of tokens associated with identified hallucination risks. Our main contributions are as follows:

- We propose NDAD, an innovative decoding approach that introduces hallucination signal to expose the model's underlying hallucination distribution and applies a negative awareness mechanism for intervention.

- We incorporate a global weight measuring the directional consistency between hallucination signal and original early-layer logits, and a local weight quantifying the likelihood of tail tokens evolving toward the mature distribution.

- We perform comprehensive experiments on a diverse set of LLMs with different configurations and scales. The experimental results indicate that NDAD reliably enhances factual accuracy across multiple tasks and benchmark datasets.

## 2 RELATED WORK

**Hallucination Mitigation.** In LLMs, hallucination refers to the generation of content that diverges from factual knowledge, and it has become a critical bottleneck for ensuring model reliability. Existing research has proposed mitigation strategies along several directions. Retrieval-based methods introduce external knowledge to calibrate factuality, such as Retrieval-Augmented Generation (RAG) (Cheng et al., 2023; Chen et al., 2024a; Fan et al., 2024; Lewis et al., 2020), or enhance attribution by applying retrieval and editing after generation to improve both factuality and traceability (Gao et al., 2022; Mishra et al., 2024). Training- and preference-based methods rely on additional supervised data or human preference signals for optimization, including Supervised Fine-Tuning (SFT) (Tian et al., 2023), Reinforcement Learning with Human Feedback (RLHF) (Ouyang et al., 2022), and Direct Preference Optimization (DPO) (Rafailov et al., 2023), thereby reducing hallucination through parameter updates. Self-evaluation-based methods do not rely on external data; instead, they improve reliability by leveraging multiple inference-time samples and incorporating techniques such as self-criticism (Saunders et al., 2023) and diversified reasoning path sampling (Wang et al., 2022). To further improve efficiency in enhancing factuality, our goal is to directly optimize the output distribution of language models, thereby strengthening their robustness.

**Intervention Decoding.** In recent years, a line of research has emerged that enhances the factuality of LLMs by intervening during the decoding time. Inference-Time Intervention (ITI) (Li et al., 2023a) identifies attention heads correlated with truthfulness during inference and shifts activations along these "truthful directions", thereby enhancing the truthfulness of generated outputs. Similarly, Activation Decoding (AD) (Chen et al., 2024b) leverages the model's internal representations by introducing an entropy-based metric of contextual activation sharpness as a decoding constraint, thereby biasing outputs toward more reliable generations. Inspired by early work on Contrastive Decoding (CD) (Li et al., 2022), which compared strong expert models against weaker amateur models to improve fluency and coherence without addressing factuality, subsequent studies extended the idea of "contrast" to the logits level. For example, Auto-Contrastive Decoding (ACD) (Gera et al., 2023) requires fine-tuning the prediction heads of earlier layers and is therefore mainly applicable to small-scale models. In contrast, Decoding by Contrasting Layers (DoLA) (Chuang et al., 2023) dynamically selects the early layer that exhibits the largest semantic divergence from the final layer, thereby suppressing erroneous tendencies in lower layers. Building upon this, Self Logits Evolution Decoding (SLED) (Zhang et al., 2024) further integrates multiple early layers through weighted combination and employs a gradient-descent procedure to guide the correction of the final logits, resulting in more robust factuality enhancement.

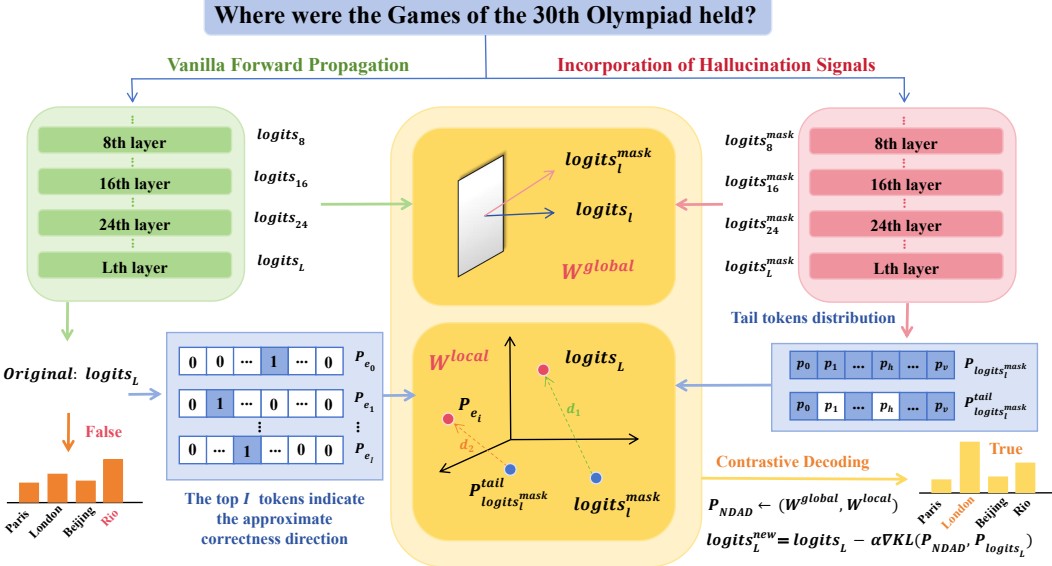

Figure 1: Overview of NDAD. To enhance factual reliability, we introduce hallucination signals to adjust the final output logits. The weights $\mathcal{W}^{\text{global}}$ and $\mathcal{W}^{\text{local}}$ jointly regulate the hallucination signals to form the latent allucination distribution $\mathcal{P}_{\text{NDAD}}$, from which the model steers its output away.

## 3 METHOD

LLMs are designed to autoregressively predict the next token given a preceding context. Formally, given an input prefix represented as $\mathbf{x}_{<t} = \{x_1, x_2, \ldots, x_{t-1}\}$, the model first converts these tokens into a sequence of embedding vectors, $\mathcal{H}_0 = \{h_0^{[1]}, h_0^{[2]}, \ldots, h_0^{[t-1]}\}$, through an embedding layer. These representations are then updated successively by a stack of $L$ transformer blocks. We denote the hidden state of the $t$-token at the $l$-th block as $h_l^{[t]} \in \mathbb{R}^{d_h}$. To generate a probability distribution over the model's vocabulary $\mathcal{V}$, a shared projection head $\psi : \mathbb{R}^{d_h} \to \mathbb{R}^d$ is applied to the hidden states. In detail, from the $l$-th layer's hidden state, the unnormalized score vector (logits) for the next token and its corresponding probability distribution are defined as:

$$P_l^{[t]} = \text{softmax}(logits_l^{[t]}), \text{ where } logits_l^{[t]} = \psi(h_l^{[t]}),\ l = 1, \cdots, L. \tag{1}$$

Typically, the logits from the final layer, $logits_L^{[t]}$, are used for decoding. However, this can lead to generations that are plausible but factually incorrect or nonsensical. To mitigate this issue, we propose NDAD, which adjusts the logits by leveraging hallucination signal, thereby improving the reliability of the generated text.

### 3.1 HALLUCINATION SIGNAL GENERATION

Unlike prior approaches such as DoLa (Chuang et al., 2023) and SLED (Zhang et al., 2024), which harness early-layer representations as a proxy for faithful evidence to reshape the final token distribution, we instead attempt to explicitly separate the hallucination signal to encourage the final output distribution to diverge from it. Intuitively, this shifts calibration from boosting positives to subtracting negatives. In this way, our method can prevent probability mass from accumulating on spurious or speculative trajectories.

Prior studies (Wu et al., 2024) have demonstrated that certain attention heads in LLMs play a critical role in preserving factuality and stabilizing generation. Once the support of these heads is weakened, the model tends to deviate from factual directions, making the decoding process more susceptible to hallucinations. Building on this insight, we exclusively mask influential heads to isolate a hallucination signal, which serves as a negative direction for contrastive decoding. To determine which heads should be masked, we adopt head importance scores from prior work (Wu et al., 2024)

to evaluate the importance of each head. Furthermore, we take into account the entropy of each layer's distribution: a lower entropy indicates that the importance is concentrated on a small subset of heads, suggesting that these heads are more influential. By integrating both head importance and layer-level entropy, we achieve a more precise selection of heads to be masked. As illustrated in Figure 2, for each block $l \in L$, following Wu et al. (2024), we first obtain a score list of $n$ heads $\{s_{l,1}, s_{l,2}, ..., s_{l,n}\}$ in this block. We then normalize the scores into a probability distribution and compute the layer entropy as follows:

$$E_l = -\sum_{i=1}^{n} p_{l,i} \log p_{l,i}, \quad p_{l,i} = \frac{s_{l,i}}{\sum_{j=1}^{n} s_{l,j}}, \quad i = 1, \ldots, n, \tag{2}$$

Here, $p_{l,i}$ denotes the normalized importance of head $i$ in layer $l$, and $E_l$ measures the uncertainty of head importance within this layer. Then we select the top $K$ layers with the lowest entropy and mask the top $x$ heads within these layers to separate hallucination signal. We represent the hallucination signal corresponding to $l$-th block as $logits_l^{mask}$. The complete algorithmic workflow can be found in Appendix Algorithm 1. After extracting these hallucination signals, the remaining question is how to leverage them to calibrate the model's final outputs.

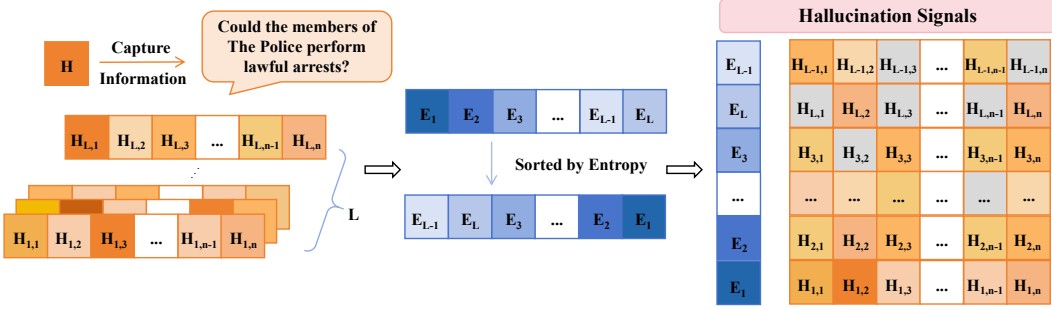

Figure 2: Hallucination signal generation. Darker colors indicate larger values, and gray cells correspond to masked attention heads.

## 3.2 DYNAMIC WEIGHTING VIA GLOBAL CONSISTENCY AND LOCAL DIVERGENCE

To exploit the identified negative direction, we propose a dynamic weighting framework that integrates both global and local perspectives.

**Global Consistency.** At the global level, we evaluate the directional consistency between the hallucination signal and the original early-layer logits from the same layer, which provides a quantitative assessment of the correlation between the original signal and the hallucination signal. Specifically, the directional consistency $c_l$ at layer $l$ is measured by computing the cosine similarity between the hallucination signal $logits_l^{mask}$ and the original logits $logits_l$ at the same layer $l$:

$$\mathcal{W}_l^{global} = \varphi(c_l), \quad c_l = \cos\_sim\left(logits_l, logits_l^{mask}\right). \tag{3}$$

where $\varphi(\cdot)$ denotes a linear mapping that scales values into the range $[0, 1]$. By measuring directional consistency, we assess the correlation between the hallucination signal $logits_l^{mask}$ and the model's original logits $logits_l$, thereby providing a quantitative basis for the referential value of hallucination signal at layer $l$. A higher consistency indicates that the signal is more closely aligned with the model's latent hallucination direction. Accordingly, the weighting scheme increases the contribution of more relevant hallucination signals.

**Local Divergence.** At the local level, we further examine the distribution of low-probability tokens. Consistent with prior studies (Chuang et al., 2023; Zhang et al., 2024), we approximate the final-layer logits $logits_L$ as the ground-truth distribution. We define the evolution trajectory from the premature to the mature state as $logits_L - logits_l^{mask}$. For the final mature layer, we further obtain the probability distribution $\mathcal{P}_{logits_L} = softmax(logits_L)$, select the top-$I$ tokens, and construct $I$ one-hot vectors $\mathcal{T} = \{\mathcal{P}_{e_1}, \mathcal{P}_{e_2}, ..., \mathcal{P}_{e_I}\}$ to serve as $I$ approximate distributions of mature, where the index of the selected token is set to $1$ and all others are set to $0$. In order to derive the hallucination distribution at

layer $l$, we begin with $\mathcal{P}_{logits_l^{\text{mask}}} = softmax(logits_l^{\text{mask}})$. Mahaut et al. (2024) suggested that low-probability tokens typically correspond to reduced factuality. Based on this observation, we define our final hallucination distribution by removing the top-$I$ tokens, which encourages the resulting distribution to approximate the negative direction more closely. In particular, by assigning a very small probability $\epsilon \to 0$ to the top-$I$ tokens in $\mathcal{P}_{logits_l^{\text{mask}}}$, we are able to derive a cleaner representation of the premature distribution, which is defined as $\mathcal{P}_{logits_l^{\text{mask}}}^{\text{tail}}$. As illustrated in Figure 1, the vector $d_1$ denotes the evolution trajectory from the premature signal $logits_l^{\text{mask}}$ at layer $l$ to the mature signal $logits_L$, while $d_2$ represents the trajectory from the hallucination distribution $P_{logits_l^{\text{mask}}}^{tail}$ toward a candidate distribution of correctness $P_{e_i}$. Both $d_1$ and $d_2$ can be interpreted as representations of the the trajectory of factual evolution, and thus we have:

$$d_1 \overset{\text{direction}}{\approx} d_2, \text{where} \quad d_1 = logits_L - logits_l^{\text{mask}}, d_2 = \nabla KL(\mathcal{P}_{logits_l^{\text{mask}}}^{\text{tail}}, \mathcal{P}_{e_i}). \tag{4}$$

Intuitively, if $d_1$ and $d_2$ are more closely aligned, it indicates that the token in $P_{logits_l^{\text{mask}}}^{tail}$ is more likely to evolve toward the mature output, and therefore a larger weight should be assigned to suppress its evolution. To quantify this evolution trajectory, we define the local weight as:

$$\mathcal{W}_{l,i}^{\text{local}} = \max\left(\text{cos\_sim}\left(logits_l^{\text{mask}} - logits_L, \mathcal{P}_{logits_l^{\text{mask}}}^{\text{tail}} - \mathcal{P}_{e_i}\right), 0\right), \quad i \in [1, I]. \tag{5}$$

After deriving both the global and local weights, we integrate them to obtain the final weight for each correctness direction within the top-$I$ tokens. Specifically, for the one-hot vector $P_{e_i}$ corresponding to the $i$-th distribution in the correctness, the final weight at layer $l$ is defined as:

$$\mathcal{W}_{l,i} = \mathcal{W}_l^{\text{global}} \mathcal{W}_{l,i}^{\text{local}}, \quad i \in [1, I]. \tag{6}$$

To better capture dominant signals and attenuate weak or noisy ones, we apply a squared transformation to the final weight scores. This operation accentuates high-confidence directions while diminishing the influence of marginal ones, thereby producing a sharper weighting distribution (Hinton et al., 2015; Müller et al., 2019; Zhang et al., 2021). Formally, the squared weight is:

$$\tilde{\mathcal{W}}_{l,i} = \left(\mathcal{W}_{l,i}\right)^2, \quad i \in [1, I] \tag{7}$$

### 3.3 NEGATIVE-DIRECTION AWARE DECODING

After introducing the global and local weighting mechanisms, we now integrate them into the overall decoding framework. NDAD leverages these weights to controllably exploit the injected hallucination signal and employs an update in the direction of gradient-descent to guide the model away from hallucination directions during generation. The following describes the specific procedure for adjusting the final-layer logits, we first perform intra-layer normalization on the obtained signals, followed by inter-layer aggregation. The squared weights $\tilde{\mathcal{W}}_{l,i}$ are normalized across the $I$ correctness directions within each layer, resulting in a layer-wise normalized distribution. Formally, the latent distribution of layer $l$ is expressed as:

$$\mathcal{P}_l = \left(\tilde{\mathcal{W}}_{l,1}, \tilde{\mathcal{W}}_{l,2}, \ldots, \tilde{\mathcal{W}}_{l,|I|}\right)/\mathcal{Z}_l, \quad \mathcal{Z}_l = \sum_{i=1}^{I} \tilde{\mathcal{W}}_{l,i} \tag{8}$$

We further apply inter-layer weighting to obtain the final NDAD distribution:

$$\mathcal{P}_{\text{NDAD}} = \sum_{l=1}^{L} \mathcal{N}_l \mathcal{P}_l, \quad \text{where} \quad \mathcal{N}_l = \frac{\mathcal{Z}_l}{\sum_{l=1}^{L} \mathcal{Z}_l}. \tag{9}$$

Here, $\mathcal{N}_l$ denotes the relative contribution of layer $l$, ensuring that the aggregation respects the proportional importance of each layer while preserving comparability across layers. By incorporating negative-direction awareness, we obtain a latent hallucination distribution $P_{\text{NDAD}}$. To suppress the generation of hallucination-prone tokens, we penalize the divergence between distribution $\mathcal{P}_{\text{NDAD}}$ and the original distribution $\mathcal{P}_{logits_L}$ using the KL divergence term. The procedure is outlined in Algorithm 2. Here, the parameter $\alpha$, referred to as the Evolution Rate and originally introduced in the (Zhang et al., 2024), controls the magnitude of adjustment applied to the logits along the gradient direction. We then obtain the final adjusted logits as shown below:

$$logits_L^{\text{new}} = logits_L - \alpha \nabla KL(\mathcal{P}_{\text{NDAD}}, \mathcal{P}_{logits_L}) \tag{10}$$

Table 1: Evaluation results of different methods on Llama models over varying datasets.

| Method | TruthfluQA(MC) | | | | Factor | CoT | |
|---|---|---|---|---|---|---|---|
| | MC1 | MC2 | MC3 | Avg. | Wiki | StrQA | GSM8K |
| Llama2-7B-base | 26.58 | 41.88 | 18.96 | 29.14 | 58.42 | 60.74 | 13.95 |
| +DoLa-low | 33.04 | 63.73 | 31.25 | 42.67 | 63.36 | 59.56 | 14.63 |
| +DoLa-high | 31.77 | 63.26 | 30.40 | 41.81 | 62.56 | 60.44 | 13.19 |
| +AD | 32.41 | 49.89 | 24.03 | 35.44 | 53.14 | 1.97 | 2.12 |
| +SLED | 34.15 | 62.57 | 31.89 | 42.87 | 67.00 | 61.27 | 14.63 |
| +NDAD | **34.39** | **62.62** | **31.98** | **43.00** | **67.30** | **61.57** | **14.86** |
| Llama2-7B-chat | 35.62 | 57.47 | 32.10 | 41.73 | 56.68 | 63.58 | 21.23 |
| +DoLa-low | 34.18 | 62.80 | 31.00 | 42.66 | 56.58 | 64.59 | 21.46 |
| +DoLa-high | 33.92 | 61.75 | 30.40 | 42.02 | 56.25 | 64.19 | 20.85 |
| +AD | 32.15 | 49.90 | 23.99 | 35.35 | 51.44 | 0.48 | 1.44 |
| +SLED | **37.09** | **63.83** | **32.96** | **44.63** | 64.80 | 64.50 | 21.53 |
| +NDAD | 36.84 | 63.42 | 32.93 | 44.40 | **65.06** | **64.67** | **21.99** |
| Llama2-13B-base | 27.59 | 43.14 | 19.53 | 30.09 | 63.79 | 65.98 | 28.81 |
| +DoLa-low | 31.57 | 62.48 | 30.41 | 41.49 | 65.70 | 66.46 | 28.51 |
| +DoLa-high | 29.38 | 63.92 | 33.62 | 42.31 | 52.84 | 60.83 | 11.90 |
| +AD | 32.15 | 49.90 | 23.99 | 35.35 | 58.18 | 2.01 | 0.00 |
| +SLED | 34.76 | 63.58 | 31.88 | 43.41 | 70.94 | 66.51 | 29.19 |
| +NDAD | **34.88** | **63.60** | **31.97** | **43.48** | **71.18** | **66.81** | **29.26** |
| Llama2-13B-chat | 36.47 | 63.06 | 32.77 | 44.10 | 61.96 | 69.65 | 36.69 |
| +DoLa-low | 34.27 | 63.27 | 31.36 | 42.97 | 60.69 | 69.48 | 35.48 |
| +DoLa-high | 31.82 | 62.55 | 31.13 | 41.83 | 54.81 | 66.51 | 33.21 |
| +AD | 32.15 | 49.90 | 23.99 | 35.35 | 56.71 | 23.14 | 0.00 |
| +SLED | 37.45 | 63.50 | 32.90 | 44.62 | 67.50 | 69.74 | 37.15 |
| +NDAD | **37.58** | **63.63** | **33.02** | **44.74** | **67.74** | **69.96** | **37.30** |

## 4 EXPERIMENTS

### 4.1 EXPERIMENTAL SETUP

**Benchmark datasets.** We evaluate our approach against strong baselines across both multiple-choice and open-ended generation tasks. For multiple-choice settings, we employ the TruthfulQA (Lin et al., 2021) dataset to measure factuality in short-answer scenarios and the FACTOR (Wiki) (Muhlgay et al., 2023) dataset to assess performance in long-paragraph contexts. For open-ended generation, we consider PopQA (Mallen et al., 2022), NQ-Open (Lee et al., 2019), and TriviaQA (Joshi et al., 2017), as well as reasoning-intensive tasks involving chain-of-thought (CoT), including StrategyQA (Geva et al., 2021) and GSM8K (Cobbe et al., 2021).

**Models and Baselines.** In our experiments, we adopt a diverse set of representative open-source LLMs, including Llama2-7B (base and chat) (Touvron et al., 2023b), Llama2-13B (base and chat) (Touvron et al., 2023b), Qwen2.5-7B-instruct (Team, 2024), Mistral-7B-instruct (Jiang et al., 2023), and Llama3-8B-instruct (Grattafiori et al., 2024). We compare the following baselines: (1) Greedy Decoding. (2) DoLA-Low (Chuang et al., 2023) subtracts the logits of the most distributionally different layer from the first half of the network from the final-layer logits. (3) DoLA-High (Chuang et al., 2023) subtracts the logits of the most distributionally different layer from the second half of the network from the final-layer logits. (4) AD (Chen et al., 2024b) uses an entropy-based measure of contextual activation sharpness to constrain decoding with the model's internal representations. (5) SLED (Zhang et al., 2024) integrates multiple early layers via weighted combination and applies a gradient-descent adjustment to refine the final logits for improved factuality.

**Metrics and Parameters.** For multiple-choice and CoT reasoning tasks, we evaluate factual accuracy following the approach in (Chuang et al., 2023). To assess correctness on TriviaQA, HotpotQA, and NQ-Open, we adopt the Exact Match (EM) metric, consistent with the protocol of (Joshi et al., 2017). The detailed parameter settings are provided in Appendix A.1.

Table 2: Evaluation results on Open-Ended generation tasks.

| Method | Llama2-7B-base | | | Llama2-7B-chat | | |
|---|---|---|---|---|---|---|
| | TriviaQA | PopQA | NQ-Open | TriviaQA | PopQA | NQ-Open |
| Greedy | 65.04 | 13.67 | 21.02 | 59.61 | 18.55 | 23.41 |
| +DoLa-low | 64.96 | 13.88 | 20.78 | 54.65 | 19.64 | 23.60 |
| +DoLa-high | 63.96 | 13.41 | 19.31 | 54.24 | 19.48 | 23.55 |
| +AD | 48.78 | 15.11 | 22.44 | 59.64 | 18.43 | 23.60 |
| +SLED | 65.10 | 25.86 | 25.96 | 59.61 | 19.98 | 23.46 |
| +NDAD | **65.21** | **26.00** | **26.26** | **59.67** | **20.13** | **23.63** |
| | Llama2-13B-base | | | Llama2-13B-chat | | |
| Greedy | 68.34 | 25.04 | 32.71 | 66.32 | 19.82 | 30.03 |
| +DoLa-low | 68.67 | 28.64 | 28.78 | 65.54 | 17.82 | 29.14 |
| +DoLa-high | 62.08 | 26.12 | 25.68 | 61.86 | 16.32 | 27.42 |
| +AD | 67.67 | 17.91 | 30.80 | 64.50 | **22.91** | **34.52** |
| +SLED | 71.47 | 30.53 | 32.52 | 66.40 | 19.84 | 29.89 |
| +NDAD | **71.66** | **30.64** | **32.88** | **66.48** | 19.85 | 30.11 |

## 4.2 EVALUATION ON DIFFERENT BENCHMARKS

**Multiple-Choices Tasks.** These tasks are designed to evaluate whether the decoding strategy can more effectively assign higher probabilities to correct answers or reasonable completions, while suppressing its preference for incorrect options. It should be noted that these tasks are essentially distribution-fitting problems, and overfitting to specific tasks often undermines the generalization capability of a decoding method. Since our goal is to enhance factuality and robustness while preserving broad applicability, even when the performance deviation is small or only marginal improvements are achieved, the results remain understandable and acceptable. We validated the effectiveness of the NDAD method through short-answer factuality tests on the TruthfulQA dataset and long-paragraph factuality tests on the FACTOR dataset. The corresponding experimental results are summarized in Table 1, and more detailed analyses are provided in Appendix B.1. Our NDAD method demonstrates strong generalization across different models and datasets, and largely achieves improvements over the baseline SLED. This suggests that the proposed decoding strategy is generally more effective at calibrating probability assignment between correct and incorrect answers.

**Chain-of-Thought Reasoning Tasks.** This task primarily focuses on evaluating how different decoding methods can be adapted to the CoT strategy to effectively handle complex reasoning problems. The detailed results can be found in Table 1. Our NDAD method consistently outperforms all baselines in decoding performance. At the same time, the limitations of AD become particularly evident on CoT datasets. AD constrains next-token probabilities by incorporating contextual entropy to enhance factuality. However, it falls short on reasoning tasks because tokens in CoT datasets exhibit strong logical dependencies, and relying solely on token-level activation entropy from the context may deviate from the original semantics. Moreover, some intermediate tokens lack contextual support and are prone to being misclassified as hallucinations, thereby impairing reasoning performance.

**Open-Ended Generation Tasks.** For open-ended tasks, we adopt TriviaQA, PopQA, and NQ-Open datasets. Our NDAD method consistently achieves further improvements over the baselines. Results are shown in Table 2. Since PopQA and NQ-Open are highly knowledge-intensive, models tend to rely more on contextual information during generation. The AD method, which is inherently designed to adjust decoding based on contextual attention, therefore shows exceptionally strong reasoning performance on the Llama-13B-chat model. However, when compared with the results on CoT tasks in Table 1, it becomes evident that AD exhibits substantial variability. Therefore, our NDAD method demonstrates the strongest robustness.

## 4.3 EVALUATION ON DIFFERENT LLMS

We further conduct experiments on a broader range of model architectures, including models from different families as well as different variants within the same family. As reported in Table 3, NDAD consistently delivers state-of-the-art results across all tested configurations, surpassing other baselines. This demonstrates that the proposed method is not only effective for a specific model class but

Table 3: Evaluation results on varying LLMs.

| Model | TruthfluQA(MC) | | | | Factor | CoT |
|---|---|---|---|---|---|---|
| | MC1 | MC2 | MC3 | Avg. | Wiki | GSM8K |
| Qwen2.5-7B-instruct | 41.00 | 64.59 | 38.17 | 47.92 | 54.54 | 84.46 |
| +DoLa-low | 36.60 | 66.03 | 34.21 | 45.61 | 56.08 | 83.02 |
| +DoLa-high | 34.64 | 2.37 | 34.51 | 23.84 | 40.85 | 76.95 |
| +SLED | 45.04 | 70.37 | 39.88 | 51.76 | 62.99 | 84.91 |
| +NDAD | **45.17** | **70.37** | **39.89** | **51.81** | **63.13** | **85.14** |
| Mistral-7B-instruct | 40.27 | 68.32 | 37.06 | 48.55 | 60.49 | 53.45 |
| +DoLa-low | 39.53 | 68.44 | 36.16 | 48.04 | 64.16 | 53.22 |
| +DoLa-high | 39.53 | 68.43 | 36.09 | 48.02 | 64.23 | 53.30 |
| +SLED | 45.41 | 71.17 | 40.27 | 52.28 | 67.53 | 53.90 |
| +NDAD | **45.53** | **71.31** | **40.46** | **52.43** | **67.70** | **54.36** |
| Llama3-8B-instruct | 38.92 | 68.16 | 36.56 | 47.88 | 59.22 | 75.97 |
| +DoLa-low | 35.74 | 65.27 | 33.60 | 44.87 | 61.32 | 75.82 |
| +DoLa-high | 35.99 | 65.04 | 33.72 | 44.92 | 61.29 | 75.51 |
| +SLED | 41.37 | 68.46 | 37.61 | 49.15 | 67.07 | 75.82 |
| +NDAD | **41.37** | **69.21** | **37.89** | **49.49** | **67.20** | **77.18** |

Table 4: Evaluation results on Llama2-70B.

| Method | Factor | GSM8K |
|---|---|---|
| Llama2-70B | 61.92 | 56.10 |
| +DoLa-Low | 74.05 | 57.01 |
| +DoLa-High | 62.53 | 38.21 |
| +SLED | 77.32 | 57.01 |
| +NDAD | **77.52** | **57.54** |

Table 5: Runtime and memory overhead on Llama2-7B-base.

| Method | Runtime (s) | Memory (MB) |
|---|---|---|
| Greedy | 1.11 | 13503.47 |
| DoLa | 1.17 | 15261.98 |
| SLED | 1.17 | 15452.88 |
| NDAD | 1.34 | 17779.01 |

also generalizes well across diverse architectures. Moreover, the performance gains are particularly pronounced on CoT datasets such as GSM8K, where NDAD exhibits substantial improvements over the baselines. This finding highlights the robustness of NDAD in handling complex reasoning tasks. Consequently, these results confirm that NDAD achieves both cross-model generality and strong robustness, making it a versatile and effective decoding strategy.

## 4.4 EVALUATION ON LARGER-SCALE LLM

To assess the viability of the method on substantially larger models, we conducted additional experiments using Llama2-70B on the Factor dataset for multiple-choice tasks and GSM8K for chain-of-thought reasoning. The results, presented in Table 4, show that the method continues to deliver strong performance on generative tasks such as GSM8K. The second-best baseline improves by 0.91%, whereas our method achieves an improvement of 1.44%, corresponding to a relative gain of 58%. For the Factor dataset, as discussed in Section 4.2, this task essentially evaluates distribution fitting, where maintaining a smooth upward trend is sufficient. These results demonstrate that the method remains effective when scaled to much larger models and exhibits strong robustness across different model sizes.

## 4.5 ABLATION STUDY

**Incorporation of Hallucination Signal.** We first demonstrate that our method indeed introduces hallucination signal into the model. To this end, we directly decode the logits obtained after masking the importance attention heads and evaluate their performance. The experimental results are shown in Figure 3. As can be observed, compared with the original decoding, performance consistently drops across different models and datasets, with the most significant decline occurring on the GSM8K dataset. This indicates that complex reasoning tasks heavily rely on the aggregation and inference of internal attention heads, and masking these heads introduces stronger hallucination signal. This observation is consistent with the analysis in Section 4.3, where our NDAD method achieves better

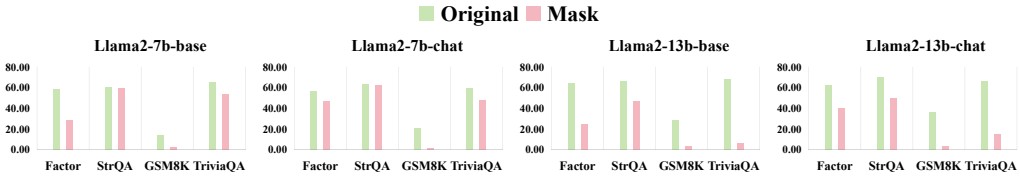

Figure 3: Results from Decoding Hallucination Signals.

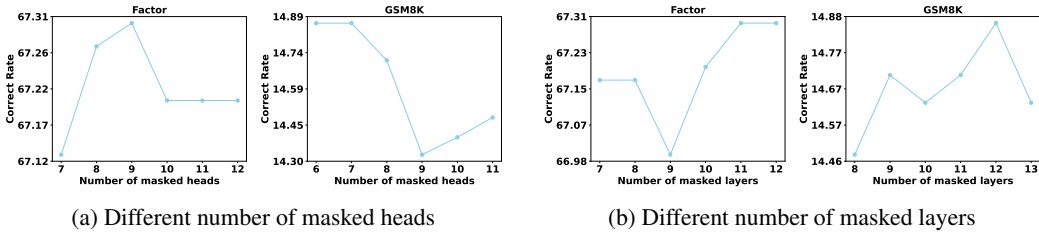

(a) Different number of masked heads      (b) Different number of masked layers

Figure 4: Different head and layer parameters on the Llama-7B-base.

results on GSM8K, suggesting that stronger hallucination signal can provide more effective leverage for enhancing the NDAD decoding strategy. Moreover, the ablation experiments in Table 6 based on random head and layer selection further support that hallucination induction guided by head importance and layer-level entropy contributes to the performance gains of NDAD.

**Importance of Head and Layer Parameters.** To effectively introduce hallucination signal, it is necessary to mask more important attention heads. Using the Llama-7B-base model as an example, we present results on the FACTOR and GSM8K datasets under different parameter settings. Figure 4a illustrates the impact on accuracy when varying the number of masked heads while keeping the number of masked layers fixed. Conversely, Figure 4b shows the effect of varying the number of masked layers while fixing the number of masked heads. Overall, the trend generally follows a rising-then-falling pattern. Notably, throughout the experiments, the range of masked heads and layers remained between [6, 13], within which the model consistently achieved relatively strong performance across both datasets. More detailed results are provided in Appendix B.3.

**Global and Local Weights.** We further analyze the effectiveness of the global and local weighting components in our method. The ablation results based on Llama2-7B-base and Llama2-13B-base are reported in Table 6, and the more comprehensive results and analyses can be found in Appendix B.2. Specifically, w/o global weight indicates removing the measurement of directional consistency between hallucination signal and the original signals, while w/o local weight corresponds to excluding the measurement of consistency between the tail-token evolution and the transition from the premature to the mature state. From the results, it is clear that both weighting mechanisms play a crucial role in enhancing the decoding performance. For example, in the case of Llama2-7B-base, removing either global or local weights leads to a drop in performance. A similar trend is observed for Llama2-13B-base, where the absence of these weights consistently reduces accuracy across all benchmarks. Importantly, the GSM8K dataset again shows the largest degradation, underscoring that complex reasoning tasks are particularly sensitive to the loss of these weighting mechanisms. These results confirm that both global and local weights contribute complementary benefits, and together they enable NDAD to achieve robust and state-of-the-art performance.

## 4.6 Computational Overhead Analysis

To evaluate the computational overhead of our method, we measured runtime and memory usage on the Llama2-7B-base model using a single GSM8K sample, and the results are presented in Table 5. As shown, the additional cost introduced by NDAD is relatively lightweight, with the primary overhead arising from the incorporation of the negative-direction signal. Consistent with existing decoding-based approaches, NDAD only modifies the logits of the final layer, requires no additional training, and does not depend on high-quality external data, giving it strong plug-and-play capability. In many real-world applications, safety and factual reliability are often more critical than achieving

Table 6: Ablation study on the effectiveness of each component in the NDAD method.

| Method | TruthfluQA(MC) | | | | Factor | CoT | |
|---|---|---|---|---|---|---|---|
| | MC1 | MC2 | MC3 | Avg. | Wiki | StrQA | GSM8K |
| Llama2-7B-base | 26.58 | 41.88 | 18.96 | 29.14 | 58.42 | 60.74 | 13.95 |
| random head | 34.15 | 62.55 | 31.91 | 42.87 | 67.17 | 61.13 | 13.95 |
| random layer | 34.15 | 62.61 | 31.84 | 42.87 | 67.10 | 61.40 | 14.71 |
| w/o global weight | 34.27 | 62.57 | 31.93 | 42.92 | 67.20 | 61.09 | 14.63 |
| w/o local weight | 33.90 | 61.13 | 31.43 | 42.15 | 67.17 | 61.44 | 14.10 |
| NDAD | **34.39** | **62.62** | **31.98** | **43.00** | **67.30** | **61.57** | **14.86** |
| Llama2-13B-base | 27.59 | 43.14 | 19.53 | 30.09 | 63.79 | 65.98 | 28.81 |
| random head | 34.88 | 63.58 | 31.94 | 43.47 | 71.04 | 66.72 | 28.13 |
| random layer | 34.76 | 63.56 | 31.91 | 43.41 | 71.01 | 66.72 | 28.66 |
| w/o global weight | 34.88 | 63.59 | 31.93 | 43.47 | 70.98 | 65.41 | 28.73 |
| w/o local weight | 34.76 | 63.57 | 31.89 | 43.41 | 70.91 | 66.07 | 27.98 |
| NDAD | **34.88** | **63.60** | **31.97** | **43.48** | **71.18** | **66.81** | **29.26** |

the absolute lowest decoding latency; thus, a moderate amount of runtime and memory overhead is generally acceptable.

## 5 CONCLUSION

We present an innovative decoding strategy NDAD, which explicitly elicits hallucination signal by masking critical attention heads and leverages them as negative directions for contrastive decoding. To controllably leverage these signals, we design a dynamic weighting mechanism: the global weight measures the directional consistency between the hallucination signal and the original early-layer logits, thereby quantifying the referential value of the current hallucination signal; the local weight characterizes the tendency of low-probability tokens to evolve toward the mature distribution. By suppressing the output probabilities of hallucination-prone tokens through gradient-descent adjustments during decoding, NDAD consistently improves factual reliability across diverse models and benchmarks, demonstrating particularly strong robustness in complex reasoning tasks. In conclusion, NDAD provides a lightweight yet effective solution for optimizing LLM decoding.

## ACKNOWLEDGMENTS

This work was supported by the Guizhou Provincial Program on Commercialization of Scientific and Technological Achievements (Qiankehezhongyindi [2025] No. 006) and Ant Group.

## ETHICAL STATEMENT

This paper presents a decoding strategy designed to improve the factual reliability of LLMs. Our research does not involve human subjects, sensitive personal data, or potentially harmful datasets. All benchmark datasets employed in our experiments are publicly available and widely used within the Natural Language Processing research community.

## REPRODUCIBILITY STATEMENT

To ensure the reproducibility of our experiments, we have provided the source codes in the supplementary materials for review. Upon acceptance of this paper, we will release the codes as open source to enable researchers to replicate and extend our experiments.

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

## A EXPERIMENTAL SETTINGS

### A.1 PARAMETER SETTINGS

For the parameters $\alpha$ in Equation 10 and the $I$ correctness distributions in Equation 5, we set the default values to $\alpha = 2$ and $I = 10$. However, due to dataset uncertainty, additional hyperparameter tuning may be required in special cases. Following the work of (Zhang et al., 2024), we test $\alpha$ from $\{0.01, 0.1, 1, 2, 5, 10\}$ and $I$ from $\{5, 10, 20, 50\}$. During the aforementioned tests, we guarantee that the chosen parameters achieve performance better than greedy decoding. On this basis, we then incorporate our hallucination signal to conduct adaptive negative-direction aware decoding. For the number of masked heads and layers used in introducing hallucination signal, we partly explained this in Section 4.5. In experiments, we usually set the range to [6, 13], which generally yields strong performance.

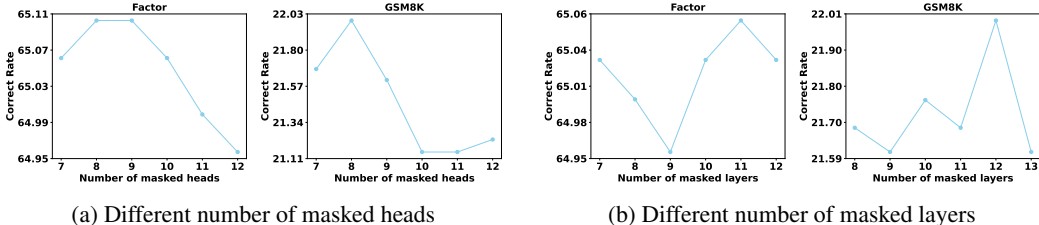

(a) Different number of masked heads  (b) Different number of masked layers

Figure 5: Different head and layer parameters on the Llama-7B-chat.

# B ADDITIONAL EXPERIMENTAL RESULTS

## B.1 EXTENDED ANALYSIS OF MULTIPLE-CHOICES TASKS

As shown in Table 1, the performance improvements of NDAD on multiple-choice tasks are slightly smaller compared to other task types. This is consistent with the discussion in Section 4.2, where multiple-choice problems essentially reduce to a logits-fitting task; as long as the model achieves stable performance without large fluctuations and delivers moderate gains, the results remain reasonable. Moreover, since the multiple-choice format inherently constrains the output space with a fixed set of candidate answers, the likelihood of hallucination is substantially reduced, leading to weaker hallucination signals and thus smaller benefits from NDAD's decoding adjustments. Nevertheless, our primary focus is on more complex open-ended generation tasks, where hallucinations are far more prevalent and where NDAD demonstrates clear advantages in suppressing hallucination-prone directions and enhancing factual reliability.

## B.2 EXTENDED ABLATION ANALYSIS

We further conducted ablation experiments on Llama2-7B-chat and Llama2-13B-chat to examine the effect of different components in NDAD, with the experimental setup summarized in Table 7.

**Hallucination Signal Induction.** During the stage of hallucination signal induction, we observed that the random selection of attention heads or layers occasionally outperformed our guided masking strategy based on head importance and layer-level entropy. This can be attributed to the inherently greedy nature of the masking strategy: although generally effective, it does not fully explore the extensive search space. Consequently, certain random configurations may fortuitously yield superior outcomes. Nonetheless, such instances are expected and do not diminish the overall effectiveness of a principled importance-guided approach.

**Global Weighting in Multiple-Choice Tasks.** For the global weighting component, the performance on Llama2-7B-chat with the TruthfulQA dataset was slightly better when the global weighting was not applied compared to the full NDAD method. As discussed in Section B.1, these multiple-choice tasks essentially reduce to a logits-fitting problem with a small set of candidate answers. Since all options are inherently more reliable than open-ended generations, the model is less vulnerable to noisy hallucinations in this setting. Consequently, assessing the reliability of hallucination signals becomes less critical, and the global weighting may even introduce unnecessary adjustments that interfere with straightforward logits alignment. By contrast, in open-ended generation tasks, where hallucination is more prevalent, the global and local weighting strategies play a much more important role in enhancing factual reliability.

## B.3 EXTENDED PARAMETER ANALYSIS

We further conducted hyperparameter experiments on Llama2-7B-chat. As shown in Figure 5, for both the number of masked attention heads and the number of masked layers, performance exhibits a general rising-then-falling trend: as the number of masked heads or layers increases, performance initially improves but declines once the masking becomes excessive. The results suggest that the optimal settings typically lie within the range of 6 to 13, where a better balance is achieved between inducing hallucination signals and preserving the original representations.

Table 7: Additional ablation study on the effectiveness of each component in the NDAD method.

| Method | TruthfluQA(MC) | | | | Factor | CoT | |
|---|---|---|---|---|---|---|---|
| | MC1 | MC2 | MC3 | Avg. | Wiki | StrQA | GSM8K |
| Llama2-7B-chat | 35.62 | 57.47 | 32.10 | 41.73 | 56.68 | 63.58 | 21.23 |
| random head | 36.84 | 63.38 | 32.65 | 44.29 | 64.93 | **64.72** | 21.15 |
| random layer | 36.47 | 62.99 | 32.59 | 44.02 | 65.00 | **64.72** | 20.62 |
| w/o global weight | **36.84** | **63.71** | **32.80** | **44.45** | 64.93 | 64.37 | 21.00 |
| w/o local weight | 36.47 | 60.82 | 32.40 | 43.23 | 64.96 | 63.58 | 20.62 |
| NDAD | 36.84 | 63.27 | 32.76 | 44.29 | **65.06** | 64.67 | **21.99** |
| Llama2-13B-chat | 36.47 | 63.06 | 32.77 | 44.10 | 61.96 | 69.65 | 36.69 |
| random head | 37.45 | 63.61 | 32.95 | 44.67 | 67.47 | 69.91 | 35.63 |
| random layer | **37.70** | **63.58** | **33.07** | **44.78** | 67.57 | 69.52 | 35.78 |
| w/o global weight | 35.62 | 63.91 | 32.49 | 44.01 | 67.60 | 69.43 | 35.71 |
| w/o local weight | 37.21 | 64.02 | 32.90 | 44.71 | 67.67 | 69.65 | 37.00 |
| NDAD | 37.58 | 63.63 | 33.02 | 44.74 | **67.74** | **69.96** | **37.30** |

## B.4 EXTENDED LINGUISTIC QUALITY EVALUATION

To assess whether NDAD introduces any degradation in linguistic quality, we conduct an additional evaluation focusing on fluency, coherence, and comprehensibility. These dimensions reflect whether the generated responses remain natural, logically organized, and easy to understand—qualities that are essential for real-world deployment but are often overlooked in factuality-oriented methods. We generate model outputs using Llama2-70B on GSM8K and obtain linguistic quality scores from the external evaluator Gemini-2.5-Pro. The results are presented in Table 8. As shown, the scores across all methods are highly consistent, and NDAD performs on par with or slightly better than existing decoding strategies, indicating that NDAD does not introduce noticeable negative effects on linguistic quality. This evaluation further demonstrates that NDAD improves factuality while preserving the naturalness and readability of generated text. Table 9 is the full evaluation prompt used for scoring with the Gemini model.

Table 8: Linguistic quality evaluation of different decoding methods using Gemini-2.5-Pro.

| Method | Fluency | Coherence | Comprehensibility |
|---|---|---|---|
| Greedy | 9.37 | 7.96 | 8.65 |
| DoLa | 9.29 | 7.91 | 8.58 |
| SLED | 9.32 | 8.02 | 8.69 |
| NDAD | 9.31 | 8.04 | 8.67 |

## C ALGORITHM OF NDAD

The entire algorithmic workflow of the NDAD method is presented in Algorithm 1 and 2.

## D CASE STUDY

Table 10 reports the results of the Llama-7B-Base model on the GSM8K dataset under different decoding strategies. The examples demonstrate that our NDAD method is more effective in eliciting factual outputs from the model.

Table 9: Prompt for Gemini-2.5-Pro.

```
You are an advanced artificial intelligence review system
specialized in evaluating the quality of model responses.  Your
task is to rate the quality from three perspectives:  fluency,
coherence, and comprehensibility.  Please strictly follow the
evaluation dimensions below to score each item (range:  0--10,
with higher scores indicating better quality).

[Evaluation Criteria]

Fluency:  Whether the sentence structure of the answer is clear
and natural, with no obvious grammatical errors, inappropriate
word usage, or issues affecting the reading experience.  Higher
scores indicate smooth language that can be read without
difficulty.

Coherence:  Whether the logical connections between parts of
the answer are tight and information flows smoothly.  Check
for jumps, breaks, contradictions, or repetition that affect
logical coherence.  Higher scores indicate clear thinking and
reasonable structure.

Comprehensibility:  Whether the answer is easy for the target
reader to understand.  Higher scores indicate clear information
delivery, easy understanding, and no ambiguity or obscure
expressions.

[Output Format]

Please output in the following JSON format:

{

"Scores for Each Dimension":  {
"Fluency":  score,
"Coherence":  score,
"Comprehensibility":  score
},

"Reason for Scoring":  Explain the reasons for scoring each
dimension, and briefly summarize the overall evaluation

}

Please validate the question and return the result in JSON
format, with no other content except the JSON.
```

---

**Algorithm 1** Hallucination Signal Induction

---

1: LLM with $L$ layers, *sequence*, following the work of (Wu et al., 2024), a original score list of $n$ attention head $\{s_{l,1}, s_{l,2}, ..., s_{l,n}\}$ in layer $l$, number of masked attention heads $x$, number of masked layer $K$.
2: **for** $l < L$ **do**
3:     Normalize scores into probability distribution: $p_{l,i} = \frac{s_{l,i}}{\sum_{j=1}^{n} s_{l,j}}, \quad i = 1, \ldots, n.$
4:     Compute attention head scores distribution entropy: $E_l = -\sum_{i=1}^{n} p_{l,i} \log p_{l,i}.$
5: **end for**
6: Obtain the set of distribution entropy $\{E_1, E_2, ..., E_L\}$.
7: Select the set $\mathcal{L}$ consisting of the $K$ layers $l$ corresponding to the lowest entropy values.
8: **for** $l \in \mathcal{L}$ **do**
9:     Set the weights of the top-$x$ scoring attention heads to $0$.
10: **end for**
11: The *sequence* into the LLM to obtain the hallucination signals $logits_l^{\text{mask}}$, where $l \leq L$.
12: **Return:** $\{logits_1^{mask}, logits_2^{mask}, ..., logits_L^{mask}\}$

---

---

**Algorithm 2** Negative-Direction Aware Decoding

---

1: **Initialization:** LLM with $L$ layers, *sequence*, $\alpha$ in Equation 10, number of correctness directions $I$, $\epsilon \to 0$, $\varphi(\cdot)$ maps values into $[0,1]$, the one-hot vectors $\mathcal{T} = \{\mathcal{P}_{e_1}, \mathcal{P}_{e_2}, ..., \mathcal{P}_{e_I}\}$ of correctness directions.
2: The *sequence* into the LLM to obtain the original logits $logits_l$ and hallucination signal $logits_l^{\text{mask}}$ given by Algorithm 1, the probabilities at each layer $l$ denoted as $\mathcal{P}_{logits_l} = softmax(logits_l)$ and $\mathcal{P}_{logits_l^{\text{mask}}} = softmax(logits_l^{\text{mask}})$, where $l \leq L$.
3: Identify the tokens with the top-$I$ largest probabilities in $\mathcal{P}_{logits_L}$ and assign the value 1 to their indices and 0 to the remaining positions.
4: Set the indices of top-$I$ largest probabilities tokens in $\mathcal{P}_{logits_l^{\text{mask}}}$ to $\epsilon$: $\mathcal{P}_{logits_l^{\text{mask}}} \to \mathcal{P}_{logits_l^{\text{mask}}}^{\text{tail}}$.
5: **for** $l < L$ **do**
6:     Compute $\mathcal{W}_l^{\text{global}} = \varphi\Big( \cos\_sim\big(logits_l, logits_l^{\text{mask}}\big) \Big).$
7:     Compute $\mathcal{W}_{l,i}^{\text{local}} = \max\Big( \cos\_sim\big(logits_l^{\text{mask}} - logits_L, \mathcal{P}_{logits_l^{\text{mask}}}^{\text{tail}} - \mathcal{P}_{e_i}\big), 0\Big), \mathcal{P}_{e_i} \in \mathcal{T}.$
8:     Calculate $\tilde{\mathcal{W}}_{l,i} = \big(\mathcal{W}_l^{\text{global}} \mathcal{W}_{l,i}^{\text{local}}\big)^2, \quad i \in [1, I].$
9: **end for**
10: Obtain the current latent distribution $\mathcal{P}_{\text{NDAD}} = \frac{\sum_{l=1}^{L} \tilde{\mathcal{W}}_{l,i}}{\sum_{l=1}^{L} \sum_{j=1}^{|I|} \tilde{\mathcal{W}}_{l,j}}$ by computing each $i \in [1, I]$ across different layers.
11: **Return:** $logits_L^{\text{new}} = logits_L - \alpha \nabla KL(\mathcal{P}_{\text{NDAD}}, \mathcal{P}_{logits_L})$

---

Table 10: Case study of Llama-7B-base on the GSM8K Dataset.

| | |
|---|---|
| **Input:** | Q: There are 15 trees in the grove. Grove workers will plant trees in the grove today. After they are done, there will be 21 trees. How many trees did the grove workers plant today? A: There are 15 trees originally. Then there were 21 trees after some more were planted. So there must have been 21 - 15 = 6. The answer is 6.
Q: If there are 3 cars in the parking lot and 2 more cars arrive, how many cars are in the parking lot? A: There are originally 3 cars. 2 more cars arrive. 3 + 2 = 5. The answer is 5.
Q: Leah had 32 chocolates and her sister had 42. If they ate 35, how many pieces do they have left in total? A: Originally, Leah had 32 chocolates. Her sister had 42. So in total they had 32 + 42 = 74. After eating 35, they had 74 - 35 = 39. The answer is 39.
Q: Jason had 20 lollipops. He gave Denny some lollipops. Now Jason has 12 lollipops. How many lollipops did Jason give to Denny? A: Jason started with 20 lollipops. Then he had 12 after giving some to Denny. So he gave Denny 20 - 12 = 8. The answer is 8.
Q: Shawn has five toys. For Christmas, he got two toys each from his mom and dad. How many toys does he have now? A: Shawn started with 5 toys. If he got 2 toys each from his mom and dad, then that is 4 more toys. 5 + 4 = 9. The answer is 9.
Q: There were nine computers in the server room. Five more computers were installed each day, from monday to thursday. How many computers are now in the server room? A: There were originally 9 computers. For each of 4 days, 5 more computers were added. So 5 * 4 = 20 computers were added. 9 + 20 is 29. The answer is 29.
Q: Michael had 58 golf balls. On tuesday, he lost 23 golf balls. On wednesday, he lost 2 more. How many golf balls did he have at the end of wednesday? A: Michael started with 58 golf balls. After losing 23 on tuesday, he had 58 - 23 = 35. After losing 2 more, he had 35 - 2 = 33 golf balls. The answer is 33.
Q: Olivia has $23. She bought five bagels for $3 each. How much money does she have left? A: Olivia had 23 dollars. 5 bagels for 3 dollars each will be 5 x 3 = 15 dollars. So she has 23 - 15 dollars left. 23 - 15 is 8. The answer is 8.
Q: Marcell and Beatrice are having a contest to see who can eat the most fruit roll-ups, so they unroll as many as they can find. Unfortunately, someone makes a mistake and Beatrice's was two roll-ups wide and 24 rolls up long while Marcell's was 3 roll-ups wide and 14 roll-ups long. If they both ate their entire amount, how many did they eat on average?
A: |
| **Dreedy** | Marcell's was 3 wide and 14 long. Beatrice's was 2 wide and 24 long. So Marcell's was 3 * 14 = 42 rolls. Beatrice's was 2 * 24 = 48 rolls. So Marcell ate 42 rolls and Beatrice ate 48 rolls. 42 + 48 = 90. The answer is 90. |
| **SLED** | Marcell's was 3 wide and 14 long. Beatrice's was 2 wide and 24 long. So Marcell's was 3 * 14 = 42 rolls. Beatrice's was 2 * 24 = 48 rolls. So Marcell ate 42 rolls and Beatrice ate 48 rolls. 42 + 48 = 90 rolls. The answer is 90. |
| **NDAD** | Marcell's was 3 roll-ups wide and 14 roll-ups long. Beatrice's was 2 roll-ups wide and 24 roll-ups long. So Marcell's was 3 * 14 = 42 roll-ups long. Beatrice's was 2 * 24 = 48 roll-ups long. So Marcell's was 42 roll-ups long and Beatrice's was 48 roll-ups long. So they both ate 42 + 48 = 90 roll-ups. 90 / 2 = 45. The answer is 45. |

