# OpenReview forum: "NDAD: Negative-Direction Aware Decoding for Large Language Models via Controllable Hallucination Signal Injection"
_ICLR.cc/2026/Conference — ICLR 2026 Poster_

### Official Review · Reviewer_ehY7 · 2025-10-30

**Soundness:** 3
**Presentation:** 3
**Contribution:** 2
**Rating:** 4
**Confidence:** 3

**Summary:**

This paper introduces Negative-Direction Aware Decoding (NDAD), a raining-free decoding strategy designed to mitigate hallucinations in LLMs. The core contribution lies in its approach of actively identifying and leveraging "hallucination signals" as a repulsive force during generation. Rather than solely promoting factual content, NDAD isolates hallucination-prone directions by strategically masking influential attention heads. The method then employs a dynamic weighting mechanism, integrating both global consistency and local divergence measures, to controllably steer the model's output away from these identified negative trajectories via a gradient-descent adjustment. Experiments demonstrate that NDAD consistently enhances factual reliability across diverse LLMs and benchmark datasets, offering a lightweight yet potent solution to a critical challenge in model deployment.

**Strengths:**

- The method is training-free and operates at inference time, making it a practical and computationally accessible approach for improving factuality compared to methods requiring model fine-tuning
- The dual-weighting scheme that considers both global consistency (alignment with layer activations) and local divergence (evolution of low-probability tokens) provides a non-trivial method for modulating the hallucination signal.
- The paper provides ablation experiments that investigate the necessity of its core components, such as the global and local weights and the importance-guided masking strategy, strengthening the justification for its specific design choices.

**Weaknesses:**

- The performance improvements over the baseline are often marginal, questioning the method's practical significance.
- The method introduces a number of new hyperparameters (number of masked heads, number of layers, top-I tokens, evolution rate α) that likely require careful tuning for different models and tasks, potentially limiting its out-of-the-box utility.
- The paper does not analyze the computational overhead or potential impact on generation latency. Furthermore, evaluation is focused on factuality, with no assessment of whether the method harms other text quality aspects like coherence or fluency.
- All models tested are in the smaller 7B-13B parameter range. The method's viability and computational cost on much larger models (e.g., 70B+) remain unproven.

**Questions:**

1. What is the computational overhead (e.g., latency increase per token) of NDAD compared to greedy decoding and the baseline?
2. Have you evaluated whether NDAD negatively impacts other text qualities, such as fluency, coherence, or stylistic appropriateness?
3. How do you expect NDAD's performance and computational cost to scale when applied to much larger models (e.g., 70B+ parameters)?

---

> ### Author Response · Authors · 2025-11-24
>
> We thank the reviewer for the detailed comments.
>
> **W1:** Regarding the improvement magnitude, it is important to note that lightweight **decoding based methods [1][2][3] enhance factuality by leveraging the model's own internal capabilities to mitigate hallucinations.** As a result, their gains are naturally smaller than approaches that rely on external retrieval or additional training. Nevertheless, when compared with other dynamic-decoding baselines, NDAD achieves clearly stronger improvements. For example, in the newly added experiments presented in Point 3 of our response, on GSM8K with Llama2-70B, the second-best baseline improves by 0.91%, while our method achieves a 1.44% gain, corresponding to a 58% relative improvement. Moreover, results already presented in the paper demonstrate that NDAD consistently yields stable gains across different models and datasets. These findings collectively indicate that the performance improvement brought by NDAD is meaningful.
>
> **W2:** Regarding the top-I tokens and evolution-rate settings, we follow the prior work [1], and the details are presented in the orginal paper **Appendix A.1**. As shown in the orginal paper **Appendix B.3 (i.e., hyperparameter sensitivity analysis)**,  the performance remains stable across a relatively broad range, with differences mainly reflected in the magnitude of improvement. In practice, we also observe stable "default values" — for example, parameters in the range of 6–13 perform well across different models and datasets. In addition, if further adjustment is needed in deployment scenarios, a very small validation set is sufficient for quickly selecting a suitable value, keeping the tuning cost very low while ensuring stability in real-world use.
>
> **W3:** We evaluate different baselines using Llama2-70B to generate the model outputs on GSM8K, and then score these outputs with the external evaluator Gemini-2.5-Pro along three dimensions—fluency, coherence, and comprehensibility. The results are shown in the table below. The scores across different methods are highly consistent, indicating that **our decoding based method does not introduce noticeable negative effects on linguistic quality**.
>
> | Llama2-70b | Fluency | Coherence | Comprehensibility |
> | :---: | :---: | :---: | :---: |
> | Greedy | 9.37 | 7.96 | 8.65 |
> | DoLa | 9.29 | 7.91 | 8.58 |
> | SLED | 9.32 | 8.02 | 8.69 |
> | NDAD | 9.31 | 8.04 | 8.67 |
>
>
>  Below is the prompt used for scoring with the Gemini model:
>
> ```plain
> You are an advanced artificial intelligence review system specialized in evaluating the quality of model responses. Your task is to rate the quality from three perspectives: fluency, coherence, and comprehensibility. Please strictly follow the evaluation dimensions below to score each item (range: 0-10, with higher scores indicating better quality).
> [Evaluation Criteria]
> Fluency: Whether the sentence structure of the answer is clear and natural, with no obvious grammatical errors, inappropriate word usage, or issues affecting the reading experience. Higher scores indicate smooth language that can be read without difficulty.
> Coherence: Whether the logical connections between parts of the answer are tight and information flows smoothly. Check for jumps, breaks, contradictions, or repetition that affect logical coherence. Higher scores indicate clear thinking and reasonable structure.
> Comprehensibility: Whether the answer is easy for the target reader to understand. Higher scores indicate clear information delivery, easy understanding, and no ambiguity or obscure expressions.
> [Output Format]
> Please output in the following JSON format:
> {
> "Scores for Each Dimension": {
> "Fluency": score,
> "Coherence": score,
> "Comprehensibility": score
> },
> "Reason for Scoring": "Explain the reasons for scoring each dimension, and briefly summarize the overall evaluation"
> }
> Please validate the question and return the result in JSON format, with no other content except the JSON.
> ```
>
> [1] SLED: Self Logits Evolution Decoding for Improving Factuality in Large Language Models.
>
> [2] DoLa: Decoding by Contrasting Layers Improves Factuality in Large Language Models.
>
> [3] In-Context Sharpness as Alerts: An Inner Representation Perspective for Hallucination Mitigation.

---

> > ### Author Response · Authors · 2025-11-24
> >
> > **W4:** To examine the method's viability on much larger models, we have conducted additional experiments using Llama2-70B on the Factor dataset (Multiple-Choice Tasks) and GSM8K (Chain-of-Thought Reasoning Tasks). The results confirm that our method maintains its performance on generative tasks (GSM8K): the second-best baseline improves by 0.91%, while our method improves by 1.44%. Compared to the second-best baseline, our relative improvement is 58%. As for the Factor dataset, as stated in Section 4.2, it is essentially a distribution-fitting task; maintaining a relatively smooth upward trend is sufficient. **These results indicate that NDAD remains effective even on much larger models, demonstrating strong robustness across model scales.**
> >
> > | 	       Llama2-70b								 | Factor | GSM8K |
> > | :---: | :---: | :---: |
> > | Greedy | 61.92 | 56.10 |
> > | Dola-Low | 74.05 | 57.01 |
> > | Dola-High | 62.53 | 38.21 |
> > | SLED | 77.32 | 57.01 |
> > | NDAD | **77.52** | **57.54** |
> >
> >
> > **Q1:** To address this comment, we conducted additional measurements on GSM8K using Llama2-7B under a single-sample setting. The results are as follows:
> >
> > | Llama2-7b | Runtime(s) | Memory(MB) |
> > | --- | --- | --- |
> > | Greedy | 1.11 | 13503.47 |
> > | Dola | 1.17 | 15261.98 |
> > | SLED | 1.17 | 15452.88 |
> > | NDAD | 1.34 | 17779.01 |
> >
> >
> > NDAD introduces a moderate amount of runtime and memory overhead, mainly due to the incorporation of the negative-direction signal. However, this negative signal enables NDAD to more effectively suppress hallucination-prone tokens, allowing it to achieve the largest factuality improvements across all benchmarks. Therefore, this small additional overhead is entirely acceptable for applications that prioritize reliability.
> >
> > **Q2:** We have already addressed this question in our response to W2, and you may refer to the explanation above.
> >
> > **Q3:** As shown in W4, NDAD continues to deliver stable factuality improvements when scaled to larger models (e.g., 70B). In terms of computational cost, NDAD operates only on the final-layer logits, so its additional computation is independent of model size. With sufficient computational resources, its decoding speed remains comparable to existing methods of this type.

---

> ### Comment · Reviewer_ehY7 · 2025-11-28
>
> Thank you authors for providing the additional experiments and all my concerns have been addressed. It is good to see that NDAD does not negatively impact text qualities, and also scales to larger models like 70B. In respect of that, I will raise my overall rating to 6.
>
> --
> update: there seems to be some issue with the review form and I cannot update my ratings at the moment.

---

> > ### Author Response · Authors · 2025-11-28
> >
> > We are delighted to hear that all your concerns have been addressed. We also appreciate your recognition and support.

---

### Official Review · Reviewer_pT8A · 2025-10-31

**Soundness:** 3
**Presentation:** 3
**Contribution:** 3
**Rating:** 6
**Confidence:** 3

**Summary:**

The following paper proposes a novel approach to hallucination reduction in Large Language Models (LLMs). It decerns a direction responsible for the hallucinations by masking out most influential attention heads responsible for the factuality. This direction is then subtracted from the original output logits of the model as a way to reduce the hallucinations in the final predictions. The paper showcases the effectiveness of the proposed method on a number of LLMs and datasets. It shows that proposed approach increases the accuracy on a number of benchmarks compared to existing hallucination reduction methods and studies the impact of the components driving the hallucinations.

**Strengths:**

+ The paper is well written, the method and the experiments  is clearly explained
+ The authors report results on a number of datasets and models - they show that proposed methods consistently outperform the existing baseline approaches.

**Weaknesses:**

**Discussion on limitations of the work**

The paper is well written but there are clearly some insights missing on the limitations of the proposed work.

1) The paper relies heavily on an existing algorithm that identifies heads responsible for factuality but it is unclear if the heads responsible for the factually are consistently the same for all tasks and sample sizes and what's the accuracy in identifying such heads. It is unclear how the number of heads and layers are chosen. It is hard to tell from Figure 4 what number of heads and layers one should choose.

2) It is also unclear how the results are aggregated for single token vs multi-token generation. When we say logits_L is this the logits for the final answer, a single token or aggregation of logits across multiple tokens ?

3) It is also unclear how we set the threshold to select top I tokens ?


Descriptions of the figures and tables.

The descriptions are quite vague and high level. It might be important to mention in the titles what are we exactly measuring e.g. Overall Accuracy or the Factuality Accuracy. Figure 1 is great but the description is very vague and it requires reading the rest of the paper to understand it.

It would we also interesting to describe the choice of combining global and local weights. Why do we use the product of W_global * W_local vs additive influence W_global + W_local

**Questions:**

See weaknesses
Table 1 shows that some of the baseline hallucination mitigation approaches worsen the accuracy. Why is that the case ? Isn't the purpose of those methods to increase the accuracy ?

---

> ### Author Response · Authors · 2025-11-24
>
> We thank the reviewer for the constructive reviews.
>
> **W1:** Our identification of critical attention heads follows the prior work [1], which has already conducted detailed analyses on these heads. That study also shows that these retrieval heads already exist in the base model, and that later extensions such as longer context training or alignment training hardly change their positions or functions.  Regarding how we select the number of heads and layers, as presented **in the orginal paper Appendix B.3**, performance remains stable within a relatively broad range, with the only difference being the degree of improvement. In practice, we also observe stable "default values" — for example, parameters in the range of 6–13 perform well across different models and datasets. In addition, if further adjustment is needed in deployment scenarios, a very small validation set is sufficient for quickly selecting a suitable value, keeping the tuning cost very low while ensuring stability in real-world use.
>
> **W2:** As written in Lines 140–141 of Section 3, "logits" in our paper refers to the logits of the next token generated at decoding time. Since autoregressive LLMs generate text token by token, extending our method to multi-token generation is straightforward: for each decoding step, we simply apply the same procedure to the logits of the current next-token prediction.
>
> **W3:** The definition and motivation of the top I tokens that represent factual directions follow the work[1], and we set the default value to 10. The details are provided in Appendix A.1 of the original paper.
>
> **W4:** In the updated PDF, we will refine the titles and descriptions of the relevant figures and tables, ensuring that their content is clearly indicated and more easily understandable for readers.
>
> Regarding why we use the product $W_{global} × W_{local}$ : these two weights evaluate two independent attributes of the same logits distribution, and therefore cannot be simply added.
>
> + $W_{global}$ measures whether the overall hallucination logits distribution has consistency and reliability, whether the distribution as a whole can serve as a meaningful negative reference signal.
> + $W_{local}$ measures whether a specific token within the distribution has a strong tendency to evolve toward hallucination.
>
> Only when the overall distribution is reliable ( high $W_{global}$) and the token itself exhibits strong hallucination risk ( high $W_{local}$) should that token receive stronger suppression. If addition were used, the final score would become large as long as one term is high (e.g., the distribution is reliable but the token is not risky, or the token is risky but the distribution is unreliable), which would amplify noise and undermine NDAD' s ability to capture hallucination directions.
>
> **Q1:** Regarding the cases where some baselines show worse accuracy: many existing contrastive decoding methods identify "non-factual" directions by comparing the distribution of early layers with that of the final layer. However, the semantics in early layers are not necessarily incorrect; they often still contain partially correct information. Treating early-layer representations as the negative direction can therefore lead to misclassification and cause the method to suppress tokens that should have been preserved.  In contrast, **our method constructs a more accurate "negative direction" by explicitly introducing hallucination-related negative signals, ensuring that the suppression primarily targets genuine hallucination tokens.** As a result, NDAD improves the model's overall performance and achieves higher accuracy across multiple benchmarks.
>
> [1] SLED: Self Logits Evolution Decoding for Improving Factuality in Large Language Models.

---

### Official Review · Reviewer_vTbm · 2025-11-01

**Soundness:** 3
**Presentation:** 2
**Contribution:** 2
**Rating:** 2
**Confidence:** 3

**Summary:**

The paper introduces a training-free method to reduce hallucination named NDAD. The method aim to reduce LLM hallucinations, its core ideas is to steer output distributions away from hallucination-prone directions. First, the method masks critical attention heads (selected based on metrics such as importance scores and entropy) to generate a "hallucination signal," then uses dynamic weighting (global weight for signal-logit alignment and local weight for low-probability token evolution) to control this signal's impact. Finally,  the output logits are modified by gradient descent to penalize the KL divergence from the negative distribution. The paper provides tests across multiple LLMs and different benchmarks, the results show that the method has similar results to baselines such as  DoLa, SLED.

**Strengths:**

1. The method presents a unique and compelling perspective by "subtracting negatives"—inducing a hallucination signal and then repelling the model from it.

2. The approach demonstrates broad applicability, validated across multiple model families (Llama, Mistral, Qwen) and sizes (7B, 13B, 8B) on a diverse set of tasks (TruthfulQA, GSM8K).

3. Rigorous studies validate the core design choices, including the efficacy of the masking strategy and the necessity of both the global and local weighting components.

**Weaknesses:**

1. My biggest concern is that the method is not practical at all. The proposed method requires a second, modified forward pass at every decoding step. In the meanwhile, the gradient-descent step is problmetic as well. This will result in inference-time latency and definitely problemetic in many inference settings(may lead to KV-cache related issues). All make the method impractical.

2. The motivation and calculation for the local weight (evolution trajectories, one-hot vectors) lack intuitive, step-by-step clarity

3. The introduction of several new parameters combined with the observed performance curves suggests the method may be highly sensitive to tuning. It can be critical for real-world deployment

4. The choice of using an arbitrary squared transformation for the final weight scores is not sufficiently justified.

5. Another concern is that even with introduction of overhead and hyper-parameters, the performance gain is very small and even worse compared with existing method

**Questions:**

see weakness

---

> ### Author Response · Authors · 2025-11-24
>
> We thank for reviewer for the dentailed comments.
>
> **W1:** We appreciate the reviewer's concern regarding practicality. In many real-world applications, however, safety and factual reliability are the primary requirements, and a small amount of additional latency is widely acceptable. **NDAD introduces only a modest overhead** **(refer to W5 response for detailed overhead analysis)**, essentially adding one extra **logits-level computation**, and similar to prior decoding based baselines [1][2][3], it only requires adjusting the logits of the final layer. Our method is fully plug-and-play, requiring no extra training and no reliance on high-quality external data—an assumption that retrieval-based or training-based approaches often cannot satisfy. Among hallucination-mitigation techniques, **this** **class of decoding based methods is generally the most practical**, whereas retrieval-based or training-based alternatives are typically much heavier in both computational and data requirements.  The concern about the gradient step and potential KV-cache issues is also misplaced. Our single-step gradient operation is applied _solely to the logits_, _without touching model parameters or intermediate states_, making the runtime cost near negligible. Moreover, NDAD can be seamlessly combined with other decoding strategies, further reinforcing its practicality and flexibility.
>
> **W2:** Previous decoding based approaches [2][3] treat the entire final layer logits distribution as the "factual" target in contrastive decoding. However, recent work [1] shows that separating out the top-k tokens from final layer logits and using their one-hot vectors as factual directions yields a more precise estimation of the factual signal. Regarding the local weight, its motivation is to suppress hallucination-prone tokens as they evolve toward the output layer, thereby improving the reliability of the decoded trajectory. The detailed calculation procedure is already provided in Section 3.2 after Line 210. If there are specific details that remain unclear, we would be happy to clarify them further.  In addition, the ablation study in Section 4.4 also demonstrates the effectiveness of the proposed local weight mechanism.
>
> **W3:** As shown in the **orginal paper Appendix B.3 (i.e., hyperparameter sensitivity analysis)**,  the performance remains stable across a relatively broad range, with differences mainly reflected in the magnitude of improvement. In practice, we also observe stable "default values" — for example, parameters in the range of 6–13 perform well across different models and datasets. In addition, if further adjustment is needed in deployment scenarios, a very small validation set is sufficient for quickly selecting a suitable value, keeping the tuning cost very low while ensuring stability in real-world use.
>
> **W4:** The motivation has already been explained in Lines 238–241. As stated in the paper, the squared transformation is used to emphasize dominant, high-confidence signals while suppressing weak or noisy ones, resulting in a sharper and more discriminative weighting distribution. This is a standard probability-sharpening operation that has been widely validated in prior work, including [4][5][6]. Recent studies such as [1] also employ similar noise-attenuation mechanisms.
>
> [1] SLED: Self Logits Evolution Decoding for Improving Factuality in Large Language Models.
>
> [2] DoLa: Decoding by Contrasting Layers Improves Factuality in Large Language Models.
>
> [3] In-Context Sharpness as Alerts: An Inner Representation Perspective for Hallucination Mitigation.
>
> [4] Distilling the knowledge in a neural network.
>
> [5] When does label smoothing help?
>
> [6] Delving deep into label smoothing.

---

> ### Author Response · Authors · 2025-11-24
>
> **W5:** To address your overhead concern, we conducted additional measurements on GSM8K using Llama2-7B under a single-sample setting. The results are as follows:
>
> | Llama2-7b | Runtime(s) | Memory(MB) |
> | :---: | :---: | :---: |
> | Greedy | 1.11 | 13503.47 |
> | DoLa | 1.17 | 15261.98 |
> | SLED | 1.17 | 15452.88 |
> | NDAD | 1.34 | 17779.01 |
>
>
> + NDAD introduces a moderate amount of runtime and memory overhead, mainly due to the incorporation of the negative-direction signal. However, this negative signal enables NDAD to more effectively suppress hallucination-prone tokens, allowing it to achieve the largest factuality improvements across all benchmarks. Therefore, this small additional overhead is entirely acceptable for applications that prioritize reliability.
> + Regarding the performance gain, it is important to note that **lightweight decoding based methods [1][2][3] enhance factuality by leveraging the model's own internal capabilities to mitigate hallucinations.** As a result, their gains are naturally smaller than approaches that rely on external retrieval or additional training. Nevertheless, when compared with other dynamic-decodng baselines, NDAD achieves clearly stronger improvements. For example, on GSM8K with Llama2-chat-7B, the next-best baseline improves by only about 0.3%, whereas our method achieves a 0.76% gain. In terms of relative improvement, this is more than twice that of the baseline and remains clearly significant. Moreover, compared with approaches that require additional training, dynamic decoding offers a much more lightweight and efficient cost–benefit trade-off.
>
> [1] SLED: Self Logits Evolution Decoding for Improving Factuality in Large Language Models.
>
> [2] DoLa: Decoding by Contrasting Layers Improves Factuality in Large Language Models.
>
> [3] In-Context Sharpness as Alerts: An Inner Representation Perspective for Hallucination Mitigation.

---

> ### Author Response · Authors · 2025-11-28
>
> Dear Reviewer vTBm,
>
> We thank you for the precious review time and comments. We have provided responses to your questions and the weakness you mentioned. We hope this can address your concerns.
>
> We would appreciate the opportunity to discuss whether your concerns have been addressed appropriately. Please let us know if you have any further questions or comments. We look forward to hearing from you soon.
>
> Best regards,
>
> Authors

---

### Official Review · Reviewer_VbaU · 2025-11-01

**Soundness:** 3
**Presentation:** 3
**Contribution:** 3
**Rating:** 6
**Confidence:** 3

**Summary:**

The paper proposes Negative-Direction Aware Decoding (NDAD), a training-free decoding strategy to reduce hallucinations in large language models. NDAD first intentionally disrupts a model’s most fact-relevant attention heads to induce a “hallucination-leaning” prediction, which reveals how the model would start to generate unreliable content. It then identifies the most dangerous parts of that signal — the token directions that are likely to grow into the final answer — and treats them as a negative direction. During actual decoding, NDAD pushes the model’s output distribution away from this negative direction via a small KL-based adjustment to the final logits. Across multiple benchmarks (factual QA, open-ended QA, and chain-of-thought math reasoning) and several popular LLMs, NDAD matches or outperforms prior decoding-time methods like SLED and Activation Decoding, without extra training data, external retrieval, or modifying model weights.

**Strengths:**

1. The perspective is new: instead of “pulling the answer toward the correct truth,” NDAD “pushes the model away from being wrong.” It first induces a hallucination signal — the direction the model tends to move toward incorrect content after targeted corruption — and then explicitly repels the final output probabilities away from that direction. This reframes the problem from boosting truthfulness to actively repulsing hallucination, which is conceptually distinct.
2. The method introduces a weighted mechanism that combines global consistency and local evolution, instead of bluntly suppressing all probabilities. After combining these two views, it applies normalization and squared weighting to emphasize high-confidence risky directions. This fine-grained control avoids the typical failure mode where aggressive down-weighting breaks fluency and makes the sentence fall apart.
3. The approach shows consistent gains across multiple tasks, and is particularly strong on tasks that require reasoning and multi-step thinking.

**Weaknesses:**

1. Runtime comparison with baselines. he method requires extra computation at inference (masking heads, computing global/local weights, doing the gradient-style correction), but the paper does not systematically report the runtime or memory overhead versus standard decoding or simpler contrastive methods.
2. Regarding the reliability and causality of the “critical attention head” selection process: Can the authors provide more fine-grained evidence, for example, whether the heads identified as critical tend to attend to high-confidence knowledge sources (such as factual spans in the context, entity mentions from the question, numerical cues, etc.) rather than merely contributing to fluency or syntax?

**Questions:**

Please refer to the weaknesses part.

---

> ### Author Response · Authors · 2025-11-24
>
> We thank the reviewer for the constructive reviews.
>
> **W1:** To address your concern regarding the runtime and memory, we conducted additional measurements on GSM8K using Llama2-7B under a single-sample setting. The results are as follows:
>
> | Llama2-7b | Runtime(s) | Memory(MB) |
> | :---: | :---: | :---: |
> | Greedy | 1.11 | 13503.47 |
> | DoLa | 1.17 | 15261.98 |
> | SLED | 1.17 | 15452.88 |
> | NDAD | 1.34 | 17779.01 |
>
>
> NDAD introduces a moderate amount of runtime and memory overhead, mainly due to the incorporation of the negative-direction signal. However, this negative signal enables NDAD to more effectively suppress hallucination-prone tokens, allowing it to achieve the largest factuality improvements across all benchmarks. Therefore, this small additional overhead is entirely acceptable for applications that prioritize reliability.
>
> **W2:** Our identification of "critical attention heads" follows prior work [1], which has already provided detailed causal analysis of these heads, including their contributions to factual recall and their stable attention patterns. Importantly, that study shows that these retrieval heads already exist in the base model and remain stable after context-length extension or alignment training.
>
> Our own ablation in Section 4.4 further confirms this: masking these heads leads to a clear drop in factual accuracy. Since NDAD directly reuses the same validated head-importance scoring and masking strategy, our method is grounded in both prior causal evidence and our empirical findings.
>
> The role of attention-head analysis in our paper is mainly foundational. NDAD's contribution lies in leveraging the hallucination-related signals induced by these heads to improve decoding and enhance factual reliability.
>
> [1] Retrieval head mechanistically explains long-context factuality.</font>

---

### Author Response · Authors · 2025-12-04

**Dear Area Chairs, Senior Area Chairs and Program Chairs,**

We sincerely appreciate your time and efforts during this challenging ICLR cycle. Below, we summarize below the main concerns raised by the four reviewers and how we have addressed them in the updated version of our work.

### 1. Computational Cost (Reviewer VbaU, vTbm, ehY7)
We evaluated the inference-time overhead of our method on Llama2-7B using GSM8K dataset. The results, provided in our response to **Reviewer VbaU W1**, show that NDAD incurs quite small extra inference time compared to baseline methods.

### 2. Additional Experiments
+ **Scaling to Larger Models (Reviewer ehY7)**

As suggested, we evaluated our method on large-scale model Llama2-70B (see response to **Reviewer ehY7 W4**). The results confirm that NDAD remains effective across various LLMs.

+ **Sensitivity Analysis (Reviewer vTbm, ehY7)**

As shown in **Appendix B.3 of the original paper**, the performance of NDAD is not very sensitive to the values of hyper-parameters. We also observed that NDAD with different hyper-parameters can consistently obtain performance improvements.

+ **Language Quality Assessment (Reviewer ehY7)**

Following the reviewer's suggestion, we evaluated response quality by employing LLM-judge approach (Gemini-2.5-Pro). As presented in response to **Reviewer ehY7 W3,** NDAD does not degrade fluency, coherence, or readability.

### 3. Remaining Clarifications
+ **Method Utility (Reviewer vTbm, ehY7)**

NDAD is a lightweight decoding-time method that enhances factuality by leveraging internal model signals. In contrast, external retrieval or additional training are resource-intensive, despite their promising performance. Moreover, NDAD also achieves substantial performance improvements over state-of-the-art dynamic decoding baselines. In response to **Reviewer ehY7 W1**, as shown in Table 4, NDAD achieves a **58% relative improvement** over the next-best baseline and exhibits consistent gains across models and datasets.

+ **Local Weight Motivation and Squared Weighting (Reviewer vTbm)**

The motivation and formulation of the local weight can be found in **Section 3.2 ("Local Divergence")**, and the squared weighting is explained in **Equation (7)**. Since no other reviewers raised concerns about these parts, we believe the overall presentation is clear. Nonetheless, we appreciate Reviewer vTbm's feedback and will further refine the exposition in future versions.



We sincerely thank all reviewers and Area Chairs for their insightful comments and efforts in this work again. We are encouraged that most reviewers maintain a positive attitude toward the contribution and potential of NDAD. We are also pleased that **Reviewer ehY7 raised the rating**, suggesting that our response sufficiently addressed the concerns. We appreciate the comments from Reviewer vTbm. Although we have not yet received his further response, we notice that his concerns largely overlap with those raised by other reviewers. Therefore, we believe that our clarifications can, to some extent, address his concerns as well.

---

### Meta-Review · Area_Chair_1BLo · 2026-01-02

**Summary:**

The submission initially received mixed reviews (2466). The main concerns are about:
1. The extra runtime or memory overhead.
2. The performance improvements are marginal.
3. There are lots of new hyperparameters, and it would be difficult to tune these hyperparameters.
4. Results on larger models are expected.
5. The presentation needs to be further improved, such as the descriptions of figures and tables are vague, and some motivations are unclear.

In the rebuttal, I think the authors have addressed most of the concerns, including more comprehensive results on extra runtime and memory overhead. The remaining concerns maybe the presentation part. Overall, I think that most of concerns have been addressed in the rebuttal, and recommend Accept.

**Reviewer Concerns:**

As mentioned, concerns about extra runtime/memory overhead analyses, marginal performance gains, and more results on larger models have been addressed.

**Reviewer Scores:**

I think most of the reviewers will maintain their ratings, as Reviewer ehY7 will increase his/her rating slightly (mentioned).

---

### Decision · Program_Chairs · 2026-01-26

Accept (Poster)